# An Antibody-Based Survey of *Toxoplasma gondii* and *Neospora caninum* Infection in Client-Owned Cats from Portugal

**DOI:** 10.3390/ani13142327

**Published:** 2023-07-17

**Authors:** Maria Aires Pereira, Carmen Nóbrega, Teresa L. Mateus, Daniela Almeida, Andreia Oliveira, Catarina Coelho, Rita Cruz, Paula Oliveira, Ana Faustino-Rocha, Maria J. Pires, João R. Mesquita, Helena Vala

**Affiliations:** 1Instituto Politécnico de Viseu, Escola Superior Agrária de Viseu, Campus Politécnico, 3504-510 Viseu, Portugal; cnobrega@esav.ipv.pt (C.N.); ccoelho@esav.ipv.pt (C.C.); rcruz@esav.ipv.pt (R.C.); hvala@esav.ipv.pt (H.V.); 2Global Health and Tropical Medicine (GHTM), Instituto de Higiene e Medicina Tropical (IHMT), Universidade Nova de Lisboa (UNL), R. da Junqueira 100, 1349-008 Lisboa, Portugal; 3CERNAS-IPV Research Centre, Instituto Politécnico de Viseu, Campus Politécnico, Repeses, 3504-510 Viseu, Portugal; 4Centre for the Research and Technology of Agro-Environmental and Biological Sciences (CITAB), University of Trás-os-Montes e Alto Douro, 5001-801 Vila Real, Portugal; pamo@utad.pt (P.O.); anafaustino@uevora.pt (A.F.-R.); joaomp@utad.pt (M.J.P.); 5CISAS—Center for Research and Development in Agrifood Systems and Sustainability, Escola Superior Agrária, Instituto Politécnico de Viana do Castelo, Rua Escola Industrial e Comercial de Nun’Àlvares, 4900-347 Viana do Castelo, Portugal; tlmateus@esa.ipvc.pt; 6Veterinary and Animal Research Centre (CECAV), Universidade de Trás-os-Montes e Alto Douro (UTAD), Associate Laboratory for Animal and Veterinary Sciences (AL4AnimalS) Quinta de Prados, 5000-801 Vila Real, Portugal; 7EpiUnit—Instituto de Saúde Pública da Universidade do Porto, Laboratory for Integrative and Translational Research in Population Health (ITR), Rua das Taipas, n° 135, 4050-091 Porto, Portugal; jrmesquita@icbas.up.pt; 8ICBAS—Institute of Biomedical Sciences Abel Salazar, University of Porto, Rua Jorge de Viterbo Ferreira 228, 4050-313 Porto, Portugal; dgomesalmeida23@gmail.com; 9Escola Superior Agrária de Ponte de Lima, Instituto Politécnico de Viana do Castelo, Rua D. Mendo Afonso, 147 Refóios do Lima, 4990-706 Ponte de Lima, Portugal; andreiasofia424@hotmail.com; 10Hospital Veterinário de Gaia, Rua Voltinha 82, 4415-369 Pedroso, Portugal; 11Department of Veterinary Sciences, University of Trás-os-Montes and Alto Douro, 5001-801 Vila Real, Portugal; 12Comprehensive Health Research Center, Department of Zootechnics, School of Sciences and Technology, University of Évora, 7004-516 Évora, Portugal

**Keywords:** toxoplasmosis, neosporosis, seroprevalence, risk factors, *Felis catus*

## Abstract

**Simple Summary:**

*Toxoplasma gondii* and *Neospora caninum* are intracellular parasites with a great impact on human and animal health, respectively. This work aims to investigate the presence of antibodies against *T. gondii* and *N. caninum* in client-owned cats from Portugal and to identify risk factors. A total of 183 domestic cats were sampled and their owners answered an online questionnaire designed to obtain background information. The overall anti-*T. gondii* and anti-*N. caninum* seroprevalences were 13.1% and 3.8%, respectively. An indoor lifestyle was identified as a significant protection factor against *T. gondii* infection, while the presence of a chronic disease and the presence of antibodies against *N. caninum* were identified as significant risk factors to *T. gondii* seroprevalence. To the best of our knowledge, this is the first serosurvey on *N. caninum* seroprevalence in cats from Portugal.

**Abstract:**

*Toxoplasma gondii* and *Neospora caninum* are obligate intracellular protozoan parasites infecting a wide range of hosts worldwide. However, information on the epidemiology of toxoplasmosis and neosporosis in cats from Portugal is limited. Thus, this study aims to evaluate anti-*T. gondii* and anti-*N. caninum* seroprevalence in client-owned cats from Portugal and to identify risk factors using a panel of well-characterized sera. A total of 183 domestic cats were sampled and screened for antibodies against *T. gondii* and *N. caninum* using commercial ELISA assays, and their owners answered an online questionnaire designed to obtain background information. The overall anti-*T. gondii* and anti-*N. caninum* seroprevalences were 13.1% (CI: 8.97–18.77) and 3.8% (CI: 1.87–7.68), respectively. Univariate analysis revealed that living strictly indoors was a significant protection factor (cOR: 0.053; CI: 0.005–0.627), and the presence of a chronic disease a significant risk factor (cOR: 3.106; CI: 1.062–9.082) to *T. gondii* seroprevalence. When performing multivariate analysis, only chronic disease (aOR: 57.527; CI: 1.7–1976.7) and seropositivity to *N. caninum* (aOR: 7.929; CI:0.8–82.9) were found to be a significant risk factor to anti-*T. gondii* antibodies. To the best of our knowledge, this is the first report of *N. caninum* seropositivity in cats from Portugal.

## 1. Introduction

*Toxoplasma gondii* and *Neospora caninum* are closely related obligate intracellular protozoan parasites of the phylum Apicomplexa [1]. Due to morphological similarities, *N. caninum* was initially misidentified as *T. gondii*. Subsequently, the two parasites were distinguished based on host preferences, clinical picture, ultrastructural morphology, and serological, immunohistochemical [2,3], genomic, and proteomic methods [4].

Both parasites have heteroxenous life cycles, with sexual development in the intestine of definitive hosts and asexual development in extraintestinal tissues of the intermediate hosts. Felids (domestic and wild) are the only known definitive hosts of *T. gondii*, but the parasite infects almost all warm-blooded vertebrates, including human populations that act as the intermediate host. To date, only domestic dogs (*Canis lupus familiaris*), coyotes (*Canis latrans*), wolves (*Canis lupus*), and dingoes (*Canis lupus dingo*) have been identified as definitive hosts of *N. caninum* [5,6,7,8]. Furthermore, *N. caninum* infects a restricted range of intermediate hosts, including cattle (*Bos taurus*), sheep (*Ovis aries*), and probably other warm-blooded animals [6,9,10,11,12,13]. At present there is no evidence that *N. caninum* can successfully infect humans [14].

The infectious stages of *Toxoplasma* and *Neospora* are tachyzoites, bradyzoites, and sporozoites. Tachyzoites are rapidly dividing stages of the parasite, capable of infecting a wide range of host cell types. They are located within a parasitophorous vacuole in the host cell cytoplasm and are the hallmark of active infection. Bradyzoites are the slowly multiplying encysted stage of both parasites and maintain chronic infection. Tachyzoite- and bradyzoite-containing cysts may be found in intermediate and definitive hosts. Sporozoites are enclosed within resistant oocysts and are disseminated in the environment by the definitive host through their feces [15,16].

Felids become infected with *T. gondii* by predating small mammals and birds, or by ingestion of raw or undercooked meat with bradyzoite-containing cysts. Queens infected for the first-time during gestation can infect offspring through the placenta or through breastfeeding [17,18]. After primoinfection, *T. gondii* sexual stages develop in the intestine of felids, which can excrete oocysts in feces for a short period of time. After elimination, oocysts become infectious after sporulation in the environment. Oocysts are very resistant and can survive, remaining infectious for months or years in soil and water [19,20,21,22]. Once seropositive, cats generally do not excrete oocysts again [16]. 

Intermediate hosts, as well as felids, harbor extraintestinal *T. gondii* stages. During the acute phase of infection, *T. gondii*-tachyzoites undergo rapid asexual multiplication into nucleated cells until the host’s humoral immune response contains parasite replication and promotes the formation of bradyzoite-containing cysts [15,23,24]. 

Clinical toxoplasmosis develops during replication and dissemination of tachyzoites. Primoinfection in pregnant women may cause health-threatening sequelae for the fetus, or even intrauterine death, and the reactivation of a latent infection in immunocompromised individuals can cause fatal encephalitis [25]. In cats, clinical toxoplasmosis is more severe after intrauterine infection, and kittens frequently develop hepatitis or cholangiohepatitis, pneumonia, and encephalitis. In adult cats, unspecific clinical signs can be observed [26]. 

In cats, *T. gondii* seroprevalence varies geographically, among countries, within different areas of a country, and within the same city [27] due to a variety of factors, such as climate, geographical region, animal age, and lifestyle [28], among others. 

Dogs play an important role in the epidemiology of *N. caninum*, representing an important risk factor for the occurrence of abortus and neonatal mortality in cattle and other intermediate hosts, through environmental contamination [29,30], although the parasite is efficiently transmitted vertically [10,31]. *N. caninum* infection was related to abortion outbreaks in dairy herds in the North of Portugal [32,33]. Seroprevalence registered in the Northern and Central regions of the country was 28.0%, being particularly high in cows with a history of abortion (46.0%) [34]. 

Regarding dogs, in the first seroprevalence study performed in Portugal, 7.9% of the domestic canids presented antibodies against *N. caninum*, but significant differences were observed among regions, with a higher prevalence in Lisboa and North regions [35]. Canine infection can result in clinical disease, characterized by paralysis that progresses to rigid contracture of the muscles, particularly in young dogs [29,36,37]. Higher *N. caninum* seroprevalences were observed in dogs with musculoskeletal and neurological signs (21.4%) [35]. 

Experimental studies demonstrated that cats are susceptible to *N. caninum* infection, developing more severe lesions if treated with corticosteroids, or infected in neonatal and prenatal periods, resembling those observed in dogs [15,38]. However, there are only a few reports of naturally acquired seropositivity to *N. caninum* among cats worldwide [27,39,40,41,42,43]. In Portugal, *N. caninum* seroprevalence in cats is unknown. 

Thus, the aim of this study was to evaluate the presence of specific anti-*T. gondii* and anti-*N. caninum* antibodies in client-owned cats from Portugal and identify risk factors for seropositivity using a panel of well-characterized sera from different geographical locations. 

## 2. Materials and Methods

### 2.1. Animal Recruiting and Sampling

Most of the samples analyzed in this work were obtained for conducting a previous study [44]. Briefly, convenience sampling was used to select veterinary centers (clinics and hospitals). Eighteen veterinary centers from mainland Portugal were invited by email to participate in this study. Veterinary centers that agreed to collaborate (8/18) received detailed instructions for sample collection and storage, informed consent, and a link to access an online questionnaire for owners. Most serum samples (65.0%) were obtained between June and August 2021, although the collection period was extended until the end of January 2022, when seven additional serum samples were obtained.

During health care visits, veterinary practitioners from collaborating veterinary centers invited cat owners to participate in the study. Cat owners who agreed to collaborate answered an online questionnaire designed to collect background information. Blood samples were collected according to veterinary norms into dry tubes and then centrifugated at 500 rcf for 10 min. Supernatants were transferred to 2 mL microtubes and stored at −20 °C before being sent to Escola Superior Agrária de Viseu (ESAV) laboratory. At ESAV, serum samples were stored frozen at −20 °C in a temperature-controlled freezer and thawed no more than 2 times prior to serological testing. 

### 2.2. Background Data Collection

A questionnaire was developed using an online platform (Google Forms^®^, Google LLC, San José, CA, USA) to collect background data. From the original questionnaire that was prepared in the Portuguese language, 17 questions (dichotomic, multiple choice and open-ended) were analyzed. The questions covered two main topics, specifically: characterization of cat owners (3 questions) and characterization of sampled cats, including cat origin, full signalment, lifestyle, prophylactic, and medical history (14 questions) (Appendix A, Table A1: questionnaire with English translation). For internal validation, the questionnaire was evaluated by the authors.

### 2.3. Detection of Anti-T. gondii and N. caninum IgG Antibodies

Serum samples were screened for antibodies against the P30 antigen of *T. gondii* using a commercial and already validated indirect multi-species ELISA (ID Screen^®^, Toxoplasmosis Indirect Multi-species, ID.Vet, Grabels, France). Testing was performed following the manufacturer’s instructions. Optical density (OD) was measured at a wavelength of 450 nm on a microplate reader MB 580 (Heales, Shenzhen Huisong Technology Development Co., Ltd., Shenzhen, China). For each sample, S/P% was calculated as follows: (OD sample-OD negative control)/(OD positive control-OD negative control) ×100, with serum samples presenting S/P% ≥ 50% being considered as positive, between 40% and 50% doubtful and ≤40% negative. 

Serum samples were screened for antibodies against *N. caninum* using a commercial competition multi-species ELISA (ID Screen^®^, Neospora caninum Competition, ID.Vet, Grabels, France), according to manufacturer’s instructions. Overnight incubation protocol was used, and optical density (OD) was measured on a microplate reader MB 580 (Heales, Shenzhen Huisong Technology Development Co., Ltd., Shenzhen, China) at a wavelength of 450 nm. For each sample, the competition percentage (S/N%) was calculated: (OD sample)/(OD negative control) ×100, with serum samples presenting S/N% ≤ 50% being considered as positive, 50% < S/N% ≤ 60% doubtful and >60% negative.

### 2.4. Data Processing and Statistical Analysis

The background data collected from Google Forms^®^ and the results of serologic analyses were downloaded in a database (Microsoft Excel 2016^®^; Microsoft Corp., Redmond, WA, USA). Statistical analysis was performed with IBM SPSS v.28.0.0.0 (IBM Corp., Armonk, NY, USA, 2020) using descriptive statistics and univariate and multivariate logistic regression. The association between the detection of anti-*T. gondii* and anti-*N. caninum* antibodies and the variables origin, breed, gender, reproductive state, age, lifestyle, cohabitants, vaccination, deworming, chronic disease and undergoing treatment, and seropositivity to *N. caninum* and *T. gondii* was evaluated using binomial logistic regression (univariate) and multinomial logistic regression (multivariate) analysis. Associations were considered significant with *p* < 0.05.

### 2.5. Ethics

The questionnaire was approved by the ethics committee of the Instituto Politécnico de Viseu (IPV), Viseu, Portugal. The blood collection was made according to the veterinary norms. Animal sampling was approved by the committee for Animal Welfare (ORBEA) of IPV. Cat owners were informed about the study and signed a written consent form.

## 3. Results

### 3.1. Sample Characterization

A total of 183 cats were sampled. Background information on the cat owners was obtained using an online questionnaire. Most cat owners were females (87.1%) and aged between 30 and 40 years (45.5%). According to the educational qualification, 28.7% of the owners stated that they had a degree, 24.8% had secondary education, and 20.8% had a master’s degree.

Serum samples were obtained from cats living in 11 of the 18 districts of mainland Portugal. Almost half of the samples (46.5%) were obtained in the Porto district, 14.9% in Évora, and 11.9% in the Braga district. According to their owners, cats were rescued from the street (62.3%), adopted from an association or official shelter (21.3%), or offered by a family member or friend (9.8%).

Most of the sampled cats were female (52.5%), non-fertile (93.4%), and European shorthair (83.6%), with a predominance of young adults, aging between 1–5 years (45.4%). Regarding animal environment, owners stated that 79.2% of the cats had an indoor lifestyle and 43.7% cohabited with other animals, namely dogs and cats.

Concerning medical prophylactic measures and health status, most cat owners stated that their cats were annually vaccinated (78.1%) and dewormed four times a year (67.8%), and 80.3% did not have chronic diseases (Table 1). 

### 3.2. Epidemiological Analysis of T. gondii and N. caninum-Seropositive Cats

The overall anti-*T. gondii* and anti-*N. caninum* seroprevalences were 13.1% (CI: 8.97–18.77) and 3.8% (CI: 1.87–7.68), respectively.

The 24 *T. gondii*-seropositive cats were from the districts of Braga (41.7%), Porto (29.2%), Évora (12.5%), Aveiro (8.3%), Setúbal (4.2%), and Viana do Castelo (4.2%). According to the Nomenclature of Territorial Units for Statistics (NUTS) II regions and considering the number of sampled cats per region, seroprevalence in the North (Viana do Castelo, Braga, Porto, Aveiro, and Viseu districts) was 13.3%, in Lisbon Metropolitan Area (Lisboa and Setúbal districts) was 10.0% and in Alentejo (Portalegre, Évora, and Beja districts) was 14.3%. 

The seven *N. caninum*-seropositive cats were from Braga (12.5%), Porto (8.3%), and Viana do Castelo (8.3%). Considering the number of sampled cats, the seroprevalence in NUTS II North region was 4.7% (Figure 1).

Most of the *T. gondii*-seropositive cats were rescued by their current owners from the street (70.8%). Regarding signalment, most *T. gondii*-seropositive cats were European shorthair (87.5%), females (3%), and non-fertile (91.7%). The age of seropositive cats was variable, but *T. gondii* antibodies were more frequent in adult cats aged between 6 and 10 years (50.0%). Concerning the environment, most seropositive cats had an indoor lifestyle (58.3%) and contact with other animals (91.7%), namely cats and dogs (37.5%). According to their owners, most seropositive cats were vaccinated (75.0%) and dewormed every 3 months (58.3%). Furthermore, most *T. gondii*-seropositive cats had no chronic diseases (62.5%) and were *N. caninum*-seronegative (91.7%).

Most *N. caninum*-seropositive cats were rescued (71.4%). Regarding signalment, all seropositive cats were European shorthair and females, and most of them were non-fertile (71.4%) and aged between 1 and 5 years (71.4%). Concerning the environment, all cats had an indoor lifestyle and contact with other animals, namely cats and dogs (85.7%). According to their owners, all *N. caninum*-seropositive cats were vaccinated and frequently dewormed (every 3 months). Most of them had no chronic diseases (85.7%) and were *T. gondii*-seronegative (71.4%) (Table 2).

### 3.3. Binary and Multinominal Logistic Regression

Univariate analysis revealed that living strictly indoors was a significant protection factor for *T. gondii* seropositivity with a *p* value of 0.02 and a crude odds ratio (cOR) of 0.053 (CI: 0.005–0.627). However, living outdoors was not found to significantly increase the risk of *T. gondii* seropositivity. The presence of a chronic disease was found to be a significant risk factor with a *p* value of 0.038 and a cOR of 3.106 (CI: 1.062–9.082) (increasing by 3.106 times the probability for being seropositive to *T. gondii*). In the univariate analysis, being positive for *N. caninum* was not found to increase the risk to *T. gondii* seropositivity. When performing multivariate analysis, only chronic disease (*p* = 0.025, aOR = 57.527 (CI: 1.7–1976.7) and seropositivity for *N. caninum* (*p* = 0.084, aOR = 7.929 (CI:0.8–82.9) were found to be significant risk factors to anti-*T. gondii* antibodies. 

Regarding *N. caninum*, significant variables were found in neither univariate nor multivariate analysis (protective or risk factors) (Table 3).

## 4. Discussion

A considerable number of studies have examined the seroprevalence and risk factor of *T. gondii* infection worldwide and some of them have compared the seroprevalence of *T. gondii* and *N. caninum* [40,41,42,43,45,46,47,48]. However, in Portugal, the number of serosurveys on *T. gondii* is limited and there are no studies on the seroprevalence of *N. caninum* in cats. 

The overall *T. gondii* and *N. caninum* seroprevalences obtained in this study were 13.1% and 3.8%, respectively. However, it must be considered that these values did not reflect mainland Portugal seroprevalences, since most samples were collected in Porto, Évora, and Braga districts. The seroprevalence of *T. gondii* was much higher than that of *N. caninum*, indicating that the cat population had more exposure in the natural environment to *T. gondii* than to *N. caninum*, as observed in previous studies worldwide [41,42,43,44,45,46,47], with some exceptions [48] 

Although tempting, the direct comparison of results obtained by different studies is a challenge, due to different sample sizes, animal sampling (age, outside access, exposure to other risk factors) and environmental factors. Still, considering the same sample, the seroprevalence obtained can be very different, depending on the characteristics of the serologic testing used. For example, *T. gondii* seroprevalences of 18.0% and 26.0% were obtained by two commercial agglutination test kits: an Indirect Hemagglutination Test (IHAT), and a Modified Agglutination Test (MAT), respectively [49]. Even when using the same serologic testing method, the lack of standardization and cut-off titers considered compromise comparison between studies [41].

The pooled global *T. gondii* seroprevalence estimated by meta-analysis in domestic cats was 35.0%, and in Europe was 43.0% [50]. However, *T. gondii* seroprevalences in domestic cats varied between 18.2% in the Netherlands and 81.3% in Poland (19.2% in Scotland, 20.8% in Greece, 32.3% in Cyprus, 34.8% in Spain, 41.0% in Norway, 47.6% in Hungary, 62.3% in Albania, and 63.1% in Estonia) [41,46,51,52,53]. 

In Portugal, *T. gondii* seroprevalences ranged between 20.5% and 44.2% [54,55,56,57]. However, considering that our sample is exclusively composed of owned cats, living mostly indoor, the low seroprevalence obtained (13.1%) is not surprising. Higher seroprevalences were obtained from stray cats from Lisbon city (44.2%) [56] and from Lisbon Metropolitan Area (24.2%) [55] using MAT but with different cut-offs (1:20 and ≥1:80, respectively), which may explain the discrepancies observed. A lower seroprevalence (20.5%) was found in domestic cats living in apartments in Lisbon city, also using MAT and considering a cut-off of 1:40 [57]. However, in the Trás-os-Montes e Alto Douro region, an overall seroprevalence of 35.8% was found in domestic cats (MAT, cut-off 1:20), but infection levels were significantly different according to cat lifestyle. Cats that had outdoors access (45.4%) presented higher seroprevalence than those living totally indoors (13.8%) [54]. 

Contrary to all *T. gondii* seroprevalence studies previously performed in Portugal, which used MAT with different cut-offs, our study was accomplished using a commercially validated indirect multispecies ELISA. The ID Screen Toxoplasmosis Indirect Multi-species uses the P30 antigen, considered the major surface antigen on the external surface of the plasma membrane of *T. gondii*. The agreement between this ELISA kit and IFAT on cat sera was high, with only 2/110 discordant results [58].

Infection of cats that usually do not have access to outdoors, as seems to be the case for most of the cats in this study, can occur through the diet, including raw meat, as well as other types of food (e.g., vegetables) contaminated with oocysts, or oocysts accidentally carried inside by their owners [57,59]. Background information collected in this study does not allow evaluating cat diet. However, most of the seropositive animals were rescued from the street, where they were exposed to a high infectious pressure [60], at least in the early stages of their lives. Vertical transmission of *T. gondii* via the placenta or their ingestion through milk are also possibilities [17,18,23,61,62].

The univariate analysis carried out in this study revealed that living strictly indoors was a significant protection factor against *T. gondii* seropositivity, but living outdoors was not found to significantly increase the risk of *T. gondii* seropositivity, probably due to the characteristics of our sampling and the reduced number of cats with an outdoor lifestyle. However, several studies point to an increased *T. gondii* seroprevalence in cats with outdoor access, which can be explained by the exposure to environmental contamination with oocysts and mainly the opportunity of outdoor cats to exert their predatory behavior and ingest potentially infected rodents and birds that are intermediate hosts of *T. gondii* [27,54,59,60]. 

The presence of a chronic disease was found to be a significant risk factor, both in univariate and multivariate analysis, increasing by three times the probability for being seropositive to *T. gondii*. Although concomitant infectious diseases, such as *Bartonella* spp., *Leishmania* spp., Feline Leukemia Virus (FeLV), and *Dirofilaria immitis* can modify the clinical outcome of *T. gondii* infection [27,63,64], the evidence that these diseases affect seropositivity is scarce, except for Feline Immunodeficiency Virus (FIV) [54,65]. To our knowledge, this study is the first one to identify the presence of infectious and non-infectious chronic diseases as a risk factor to *T. gondii* seropositivity. Indeed, chronic diseases, especially if induce immunosuppression, can render cats more susceptible to *T. gondii* infection. Furthermore, immunosuppression conditions can lead to reactivation of tissue cysts in chronically infected cats with subsequent release of bradyzoites, resulting in increased antigenemia and stimulation of specific humoral immune response [66]. However, the association between the presence of chronic disease and seropositivity to *T. gondii* may be a consequence of greater veterinary surveillance. In fact, all cats in this study were sampled during a veterinary visit, which reflects their owners’ concern with veterinary care.

Regarding *N. caninum*, the low seroprevalence (3.8%) found in this study may be explained by the resistance of cats to oral infection [9]. Furthermore, the low level of natural seroconversion observed in cats is in line with the seroprevalence registered in dogs (7.9%) in Portugal [35]. Although convenience sampling was carried out, with a greater representation of the districts located in the North of the country, the geographical distribution of *N. caninum*-seropositive cats shown here is in accordance with previous findings on canine [35] and bovine seroprevalence [28,34], underlining the role of dogs and cows in the life cycle of the parasite.

In Hungary, a *N. caninum* seroprevalence of 0.6% (titer 1:40) was found in a sample of urban, suburban, and rural cats, employing Indirect Fluorescent Test (IFAT) [41]. The low levels of seroconversion observed in Hungary contrast with the results obtained in Albania, where 10.3% of free-rooming cats from suburban areas were found positive with IFAT with a titer ≥ 1:100 [46]. The characteristics of sampled cats may justify the discrepancy observed in these seroprevalence. Higher seroprevalences (24.8%) were observed in stray cats from North Italy with *Neospora* Agglutination Test (NAT) considering titers ≥ 1:80 [39]. In this case, the differences may be partly explained by the higher sensitivity but lower specificity of the NAT when compared to the IFAT in the serodiagnosis of *N. caninum* infection [67]. Seroprevalence to *N. caninum* in feral cats from Majorca (Spain) assayed using a commercial competitive inhibition enzyme-linked immunosorbent assay (cELISA) and confirmed with IFAT (cut-off ≥ 1:50), considered the reference test for neosporosis, was 6.8% [42]. Surprisingly, Sedlák et al. (2014) [68] obtained seroprevalences of 33.0% and 3.9% with cELISA and IFAT, respectively, in domestic cats from different parts of the Czech Republic. In our study, samples were assayed using a commercial cELISA validated for the detection of antibodies against *N. caninum* in serum and plasma from ruminants, dogs, and other susceptible species, and had already been used in other serological studies, revealing a good inter-rater agreement (Kappa value) with *N. caninum* immunoblot [69,70]. 

How cats become naturally infected with *N. caninum* is not fully understood [29]. Experimental studies have proved *N. caninum* transplacental transmission during the acute and chronic stages of infection [38]. Although the ingestion of infected tissues is the most likely source of infection for carnivores, viable *N. caninum* parasites has not been isolated from potential cat prey, such as birds and small rodents [29]. Despite that, *N. caninum* DNA has been frequently demonstrated in neural and extra neural tissues of mice and rats [39,71,72] and bird tissues [73,74,75,76]. 

Considering the background information collected from the sampled cats, the source of infection is hard to establish. However, we can consider the possibility of vertical transmission, since most *N. caninum*-seropositive cats were rescued and probably come from litters born on the street, where exposure of mothers to sources of infection is probably higher. However, cats may become infected postpartum, while still on the street, or after being rescued by their current owners. In the latter case and considering that most seropositive cats had an indoor lifestyle, we can raise the possibility of infection through the ingestion of contaminated meat, occasionally offered by the owners [57]. As most seropositive cats co-habited with dogs, the final host of *N. caninum* [9], the possibility of environmental contamination and ingestion of oocysts cannot be discarded, as suggested by Hornok et al. (2008) [41].

The characterization of *N. caninum*-seropositive cats was very similar to that of *T. gondii*-seropositive animals, except for age. Cats with detectable antibodies against *N. caninum* were younger than *T. gondii*-seropositive cats. This finding may be explained by the ability of *N. caninum* to be vertically transmitted [4], from the queen to their offspring, while *T. gondii* seropositivity increases with age, in line with the assumption that infection is lifelong [60] and can be transmitted post-natally [27].

The association between seroprevalence of both *N. caninum* and *T. gondii* has previously been documented [40]. In our study, multivariate analysis identified *N. caninum* seroprevalence as a significant risk factor to *T. gondii* seroprevalence. The similarities between these protozoa and the epidemiological role of rodents and birds in the biological cycles of both protozoa may explain this finding [41].

## 5. Conclusions

In conclusion, our data report the exposure of client-owned cats from Portugal to *T. gondii* and *N. caninum* and identified the presence of a chronic disease and the seropositivity to *N. caninum* as significant risk factors to *T. gondii* seroprevalence. A strict indoor lifestyle appeared to be a significant protection factor to *T. gondii* seroprevalence. While the role of cats as definitive hosts of *T. gondii* is well defined, the significance of cats in the epidemiology of neosporosis still needs to be outlined. To the best of our knowledge, this is the first serosurvey on *N. caninum* seroprevalence in cats from Portugal.

## Figures and Tables

**Figure 1 animals-13-02327-f001:**
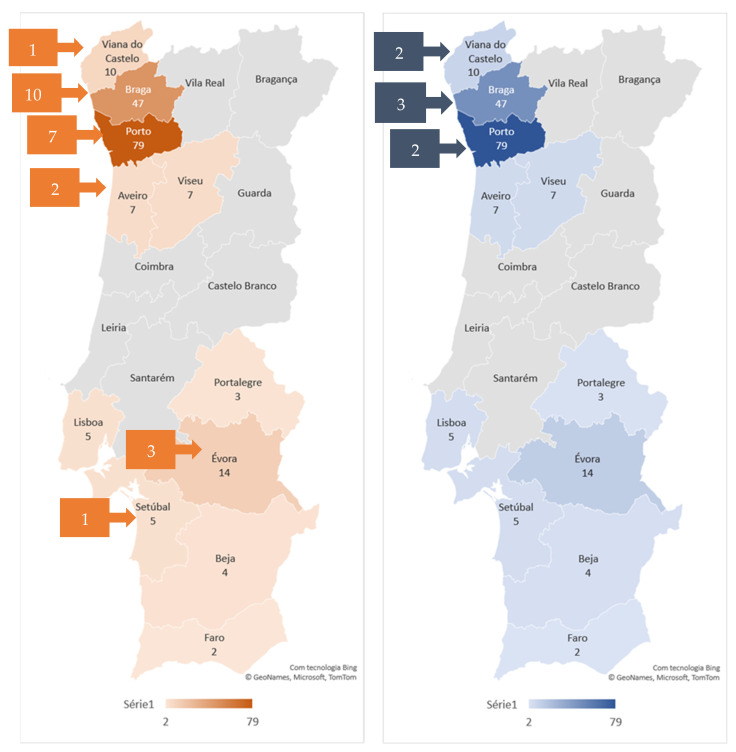
Geographical distribution of *T. gondii* (on the **left**) and *N. caninum*-seropositive (on the **right**) cats. The number of sampled cats by district is shown by the color gradient, as indicated in the legend. The number of seropositive cats by district is presented in the squares.

**Table 1 animals-13-02327-t001:** Characterization of sampled cats, including origin, residence district, signalment, environment, and clinical status.

Variables/Category	Cats Tested (n)	Relative Distribution (%)	*T. gondii* Seropositive		*N. caninum* Seropositive	
			(n)	(%)	(n)	(%)
Origin						
Rescued	114	62.3	17	14.9	5	4.4
Adopted	39	21.3	3	7.7	1	2.6
Offered	18	9.8	2	11.1	0	0
Bought	6	3.3	0	0	0	0
Other	6	3.3	2	11.1	1	16.6
District of residence						
Porto	79	43.2	7	8.9	2	2.5
Braga	47	25.7	10	21.3	3	6.4
Évora	14	7.7	3	21.4	0	0
Viana do Castelo	10	5.5	1	10	2	20.0
Aveiro	7	3.8	2	28.6	0	0
Viseu	7	3.8	0	0	0	0
Lisboa	5	2.7	0	0	0	0
Setúbal	5	2.7	1	20.0	0	0
Beja	4	2.2	0	0	0	0
Portalegre	3	1.6	0	0	0	0
Faro	2	1.1	0	0	0	0
Sex						
Male	87	47.5	10	11.5	0	0
Female	96	52.5	14	14.6	7	7.3
Reproductive status						
Fertile	12	6.6	2	16.7	2	16.7
Non-fertile	171	93.4	22	12.9	5	3.0
Age						
<1	11	6.0	0	0	0	0
1–5	83	45.4	9	10.8	5	6.0
5–10	70	38.3	12	1.4	2	2.9
>10	19	10.4	3	15.8	0	0
Breed						
European shorthair	153	83.6	21	13.7	7	4.6
Siamese	10	5.5	1	10.0	0	0
Persian	8	4.4	1	12.5	0	0
Scottish fold	2	1.1	0	0	0	0
Norwegian Forest cat	1	0.5	1	100	0	0
Other	9	4.9	0	0	0	0
Lifestyle						
Indoor	145	79.2	14	9.6	7	4.8
Indoor/outdoor	35	19.1	8	22.9	0	0
Outdoor	3	1.6	2	66.7	0	0
Co-habitants						
No	16	8.7	2	12.5	0	0
Dog and cat	80	43.7	9	11.3	6	7.5
Cat	51	27.9	8	15.7	1	2
Dog	15	8.2	2	13.3	0	0
Other animals/associations	21	11.5	2	9.5	0	0
Vaccination						
Yes	143	78.1	18	12.6	7	4.9
No	40	21.9	6	15.0	0	0
Deworming						
Every 3 months	124	67.8	14	11.3	7	5.6
Every 6 months	29	15.8	5	17.2	0	0
Annual	17	9.3	1	5.9	0	0
Occasionally	9	4.9	2	22.2	0	0
No	4	2.2	2	50.0	0	0
Chronic disease						
No	147	80.3	15	10.2	6	4.1
Yes	24	13.1	6	25.0	1	4.2
Don’t know	12	6.6	3	25.0	0	0
Type of chronic disease						
Kidney/liver disease	6	3.3	1	16.7	0	0
Retroviral infection	5	2.7	2	40.0	0	0
Heart disease	4	2.2	1	25.0	0	0
Gingivostomatitis	2	1.1	0	0	1	50.0
Hypothyroidism	1	0.5	1	100	0	0
Other	6	3.3	1	16.7	0	0
Medication						
Food supplements	6	3.3	1	16.7	1	16.7
Therapeutic commercial food	2	1.1	0	0	0	0
NSAD	1	0.5	1	100	0	0
SAD	1	0.5	0	0	0	0
Anti-*T. gondii*						
Positive	24	13.1	-	-	2	8.3
Negative	159	86.9	-	-	5	3.1
Anti-*N. caninum*						
Positive	7	3.8	2	28.6	-	
Negative	176	96.2	22	12.5	-	

NSAD—Non-steroidal anti-inflammatory drugs; SAD—Steroidal anti-inflammatory drugs.

**Table 2 animals-13-02327-t002:** Epidemiological characterization of *T. gondii* and *N. caninum*-seropositive cats.

	*T. gondii* Seropositive		*N. caninum* Seropositive	
	n = 24	%	n = 7	%
Origin				
Rescued	17	70.8	5	71.4
Adopted	3	12.5	1	14.3
Offered	2	8.3	0	0
Bought	0	0	0	0
Other	2	8.3	1	14.3
District of residence				
Porto	7	29.2	2	28.6
Braga	10	41.7	3	42.9
Évora	3	12.5	0	0
Viana do Castelo	1	4.2	2	28.6
Aveiro	2	8.3	0	0
Viseu	0	0	0	0
Lisboa	0	0	0	0
Setúbal	1	4.2	0	0
Beja	0	0	0	0
Portalegre	0	0	0	0
Faro	0	0	0	0
Sex				
Male	10	41.7	0	0
Female	13	54.2	7	100
Reproductive status				
Fertile	2	8.3	2	28.6
Non-fertile	22	91.7	5	71.4
Age				
<1	0	0	0	0
1–5	9	37.5	5	71.4
5–10	12	50.0	2	28.6
>10	3	12.5	0	0
Breed				
European shorthair	21	87.5	7	100
Siamese	1	4.2	0	0
Persian	1	4.2	0	0
Scottish fold	0	0	0	0
Norwegian Forest cat	1	4.2	0	0
Other	0	0	0	0
Lifestyle				
Indoor	14	58.3	7	100
Indoor/outdoor	8	33.3	0	0
Outdoor	2	8.3	0	0
Co-habitants				
No	2	8.3	0	0
Dog and cat	9	37.5	6	85.7
Cat	8	33.3	1	14.3
Dog	2	8.3	0	0
Other animals/associations	3	12.5	0	0
Vaccination				
Yes	18	75.0	7	100
No	6	25.0	0	0
Deworming				
Every 3 months	14	58.3	7	100
Every 6 months	5	20.8	0	0
Annual	1	4.1	0	0
Occasionally	2	8.3	0	0
No	2	8.3	0	0
Chronic disease				
No	15	62.5	6	85.7
Yes	6	25.0	1	14.3
Don’t know	3	12.5	0	0
Type of chronic disease				
Kidney/liver disease	1	4.2	0	0
Retroviral infection	2	8.3	0	0
Heart disease	1	4.2	0	0
Gingivostomatitis	0	0	1	14.3
Hypothyroidism	1	4.2	0	0
Other	1	4.2	0	0
Medication				
Food supplements	0	0	1	14.3
Therapeutic commercial food	0	0	0	0
NSAD	1	4.2	0	0
SAD	0	0	0	0
Anti-*T. gondii*				
Positive	-	-	2	28.6
Negative	-	-	5	71.4
Anti-*N. caninum*				
Positive	2	8.3	-	-
Negative	22	91.7	-	-

NSAD—Non-steroidal anti-inflammatory drugs; SAD—Steroidal anti-inflammatory drugs.

**Table 3 animals-13-02327-t003:** Binary and multinominal logistic regression analysis of *T. gondii* and *N. caninum* seropositivity in client-owned cats from Portugal.

Variable	Univariate Analysis cOR (95% CI)/*p* Value	Multivariate Analysis aOR (95% CI)/*p* Value
Lifestyle		
Indoor	0.053 (0.005–0.627)/0.02	0.096 (0–20.734)/0.393
Indoor/outdoor	0.148 (0.012–1.854)/0.139	3.499 (0.617–19.831)/0.157
Outdoor	Ref.	Ref.
Chronic Disease		
Yes	3.106 (1.062–9.082)/0.038	57.527 (0.056–1976.686)/0.025
Unknown	2.640 (0.653–10.667)/0.173	5.220 (0.056–487.669)/0.475
No	Ref.	Ref.
Seropositivity to *N. caninum*		
Positive	2.8 (0.512–15.321)/0.235	7.929 (0.759–82.878)/0.084
Negative	Ref.	Ref.

cOR: crude odds ratio; aOR: adjusted odds ratio; Ref.: variables reference level; CI: confidence interval.

## Data Availability

Data available on request due to privacy and ethical restrictions.

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
