# Peer review of "An Antibody-Based Survey of Toxoplasma gondii and Neospora caninum Infection in Client-Owned Cats from Portugal"

_animals, 2023, doi:10.3390/ani13142327_

Round 1
Author Response
Reviewer 1
The manuscript named: “AN ANTIBODY-BASED SURVEY OF TOXOPLASMA GONDII AND NEOSPORA CANINUM INFECTION IN CLIENT-OWNED CATS FROM PORTUGAL” is a important contribution to the epidemiology of this two parasites in Portugal. This paper describes, for the first time in the country that seroprevalence of Neospora in cats are presented. The manuscript is well written and is worthy publishing after addressing minor revisions such as:
Materials and Methods Line 186 to 190 – This is information is related to ethical approval not Data processing and statistical analysis.
R: The subtitle has been changed. Thank you for noticing (line 194).
Results Table 1 – Must be reeding in order to have a complet word in one line (exemple reprodutive)
R: Table has been improved. Thank you!
From line to 264 to 269 - this information belongs to section Material and methods, data processing and statistical analysis (2.4)
R: As suggested, statistical methods were removed from results and “Data processing and statistical analysis” subsection was improved.
Thank you for your careful review.
Reviewer 2 Report
It was interesting to read the article; however several main issues need to be corrected before be accepted.
Material and methods:
- To explain the categorization of cats. Eg. in line 205 you say "young adults", but, how old are considered to be young adult cats?
Results
- Lines 194-197. Data referring to the owner are indicated, but some association relating to the owner and infection by T. gondii and/or Neospora has been studied?
- Table 1 is confusing since the seroprevalence of infection by T. gondii or Neospora is calculated in relation to the total number of cats, not the number of cats in each group. Clarify it.
- Data in section 3.2. is not well understood because in line 228 it seems that the Neospora seroprevalence is 2/7 and in the Figure1 is 2/79. Clarify it.
- Figure 1 is confusing, apparently the Toxoplasma seroprevalence is higher in Braga than Porto, but in the Figure it is not clear.
- 3.2. and 3.3 are epidemiological variables analyzed. So, The title of 3.3. "3.3. Characterization of T. gondii and N. caninum seropositive cats" is not correct. Modify it. On the other hand, it is necesary to inclue p-value. A table could be designed with the data from 3.2.; 3.3; 3.4., including the p-value
Author Response
Reviewer 2
It was interesting to read the article; however several main issues need to be corrected before be accepted.
Material and methods:
- To explain the categorization of cats. Eg. in line 205 you say "young adults", but, how old are considered to be young adult cats?
R: The expression young adults was explained.
Results
- Lines 194-197. Data referring to the owner are indicated, but some association relating to the owner and infection by T. gondii and/or Neospora has been studied?
R: Yes, statistical analysis has been done, but no association was found between these variables.
- Table 1 is confusing since the seroprevalence of infection by T. gondii or Neospora is calculated in relation to the total number of cats, not the number of cats in each group. Clarify it.
R: The table was readapted to clarify your question. We have calculated the seroprevalence in relation to each group to simplify table interpretation.
- Data in section 3.2. is not well understood because in line 228 it seems that the Neospora seroprevalence is 2/7 and in the Figure1 is 2/79. Clarify it.
R: The sentence was rephrased: “The 24 T. gondii seropositive cats were from the districts of Braga (10/24; 41.7%), ….. The seven N. caninum seropositive cats were from Braga (3/7; 12.5%), …”. In addition, a table with seropositive cat characterization was constructed. We think that these improvements will facilitate the understanding of the results.
- Figure 1 is confusing, apparently the Toxoplasma seroprevalence is higher in Braga than Porto, but in the Figure it is not clear.
R: Perhaps the confusion results from the color gradient used in the map. The color gradient, refers to the number of sampled cats by district, and not to seroprevalences. Legend includes this information.
3.2. and 3.3 are epidemiological variables analyzed. So, The title of 3.3. "3.3. Characterization of T. gondii and N. caninum seropositive cats" is not correct. Modify it. On the other hand, it is necesary to inclue p-value. A table could be designed with the data from 3.2.; 3.3; 3.4., including the p-value
R: The title has been changed accordingly with your suggestion: “3.2: Epidemiological analysis of T. gondii and N. caninum seropositive cats”. Two new tables were constructed: table 2 that includes the characterization of seropositive cats and table 3 including statistically significant associations in logistic regression. We hope that these two tables will help the reader to understand the results.
Thank you for your careful review!
Reviewer 3 Report
This study describes a convenience sample of client-owned cats in Portugal, originally collected to study SARS-CoV-2 infection, tested for antibodies to T. gondii and N. caninum. I appreciate that the authors did not attempt to extrapolate the findings of their study to all cats in Portugal, which would not be appropriate given the sampling methods.
Minor revisions:
Minor suggested edit in line 151: “A questionnaire was developed using an online platform (Google Forms) to collect data from on each cat.” It’s hard ask questions of cats. In my experience they’re not very talkative.
It’s curious to note that chronic disease was a “risk factor” for T. gondii seropositivity, which may be more reflective of the cat owners who take their cats to the veterinary clinic, rather than a true association (i.e., more cats who go to the vet, and thus ended up in your sample, may have a chronic disease as opposed to healthy cats whose owners may not perceive a need for routine veterinary care). The authors did explain this finding in terms of chronic diseases inducing immunosuppression, but it would be good to also note how the sample may reflect in this finding.
Author Response
Reviewer 3
This study describes a convenience sample of client-owned cats in Portugal, originally collected to study SARS-CoV-2 infection, tested for antibodies to T. gondii and N. caninum. I appreciate that the authors did not attempt to extrapolate the findings of their study to all cats in Portugal, which would not be appropriate given the sampling methods.
Minor revisions:
Minor suggested edit in line 151: “A questionnaire was developed using an online platform (Google Forms) to collect data from on each cat.” It’s hard ask questions of cats. In my experience they’re not very talkative.
R: Thank you for noticing!
It’s curious to note that chronic disease was a “risk factor” for T. gondii seropositivity, which may be more reflective of the cat owners who take their cats to the veterinary clinic, rather than a true association (i.e., more cats who go to the vet, and thus ended up in your sample, may have a chronic disease as opposed to healthy cats whose owners may not perceive a need for routine veterinary care). The authors did explain this finding in terms of chronic diseases inducing immunosuppression, but it would be good to also note how the sample may reflect in this finding.
R: The authors are grateful for the reflection of reviewer 3. The discussion has been amended to reflect this point of view (lines 407-411).
Thank you for your careful review!
Reviewer 4 Report
The manuscript “An antibody-based survey of Toxoplasma gondii and Neospora caninum infection in client-owned cats from Portugal” aims to investigate the presence of antibodies against T. gondii and N. caninum in 183 client-owned cats from Portugal and to identify risk factors. The manuscript is well written, and should be of great interest to the readers. The work has limitations, such as a selected sampling (client-ownerd cats) influencing the % seroprevalence and these limitations are however described by the authors in the discussion section. Moreover, this manuscript represents a first survey of seroprevalence of N. caninum in cats in portugal, which may be of great interest to readers.
I suggest some changes to be made:
- in the 'results' section, insert a table to summarise the data obtained.
-the discussion section is really excessive, it should be summarised, leaving the salient points.
Finally, once these changes are made, I recommend that the article be accepted for publication in this journal.
Author Response
Reviewer 4
The manuscript “An antibody-based survey of Toxoplasma gondii and Neospora caninum infection in client-owned cats from Portugal” aims to investigate the presence of antibodies against T. gondii and N. caninum in 183 client-owned cats from Portugal and to identify risk factors. The manuscript is well written, and should be of great interest to the readers. The work has limitations, such as a selected sampling (client-ownerd cats) influencing the % seroprevalence and these limitations are however described by the authors in the discussion section. Moreover, this manuscript represents a first survey of seroprevalence of N. caninum in cats in portugal, which may be of great interest to readers.
I suggest some changes to be made:
- in the 'results' section, insert a table to summarise the data obtained.
R: Tables 2 and 3 were constructed to summarize results.
-the discussion section is really excessive, it should be summarised, leaving the salient points.
R: The discussion was summarized as suggested.
Finally, once these changes are made, I recommend that the article be accepted for publication in this journal.
R: Thank you for your review.
Round 2
Reviewer 2 Report
Thank you for your comments
I think it is ok n the present form.